# Fast-RRT: A RRT-Based Optimal Path Finding Method

**Zhenping Wu, Zhijun Meng \*, Wenlong Zhao**  **and Zhe Wu**

School of Aeronautic Science and Engineering, Beihang University, Beijing 100191, China;
wuzp0423@buaa.edu.cn (Z.W.); zhaowenlong@buaa.edu.cn (W.Z.); wuzhe@buaa.edu.cn (Z.W.)
\* Correspondence: mengzhijun@buaa.edu.cn

**Abstract:** As a sampling-based pathfinding algorithm, Rapidly Exploring Random Trees (RRT) has been widely used in motion planning problems due to the ability to find a feasible path quickly. However, the RRT algorithm still has several shortcomings, such as the large variance in the search time, poor performance in narrow channel scenarios, and being far from the optimal path. In this paper, we propose a new RRT-based path find algorithm, Fast-RRT, to find a near-optimal path quickly. The Fast-RRT algorithm consists of two modules, including Improved RRT and Fast-Optimal. The former is aims to quickly and stably find an initial path, and the latter is to merge multiple initial paths to obtain a near-optimal path. Compared with the RRT algorithm, Fast-RRT shows the following improvements: (1) A Fast-Sampling strategy that only samples in the unreached space of the random tree was introduced to improve the search speed and algorithm stability; (2) A Random Steering strategy expansion strategy was proposed to solve the problem of poor performance in narrow channel scenarios; (3) By fusion and adjustment of paths, a near-optimal path can be faster found by Fast-RRT, 20 times faster than the RRT\* algorithm. Owing to these merits, our proposed Fast-RRT outperforms RRT and RRT\* in both speed and stability during experiments.

**Keywords:** path planning; RRT; random expansion; path optimization

## 1. Introduction

Scientifically, motion planning refers to finding a continuous feasible path, which starts in the initial state and ends in the target state. Motion planning plays a key role in many applications such as unmanned aerial vehicles (UAV), self-driving cars, and mobile robots, thus greatly emerging the development of motion planning algorithms.

In the past few decades, many motion planning algorithms have been proposed. One of the important categories is graph-based methods such as Dijkstra's algorithm [1] and A\* algorithm [2]. They discretize the state space of the motion planning problem into a graph structure and then find a feasible path by using graph search methods. Among them, Dijkstra's algorithm is a breadth-first search algorithm, which can find an optimal path. While the A\* algorithm introduces a heuristic function and accelerates the search speed of the D\* algorithm. Extensive efforts were made to improve the performance of the A\* algorithm. For instance, the Jump Point Search algorithm can speed up the A\* algorithm by order of magnitude [3]. Liu et al. [4] further extended the Jump Point Search algorithm from 2D to a 3D environment. By virtue of introducing dynamic constraints, the Hybrid A\* [5] can generate smooth paths to satisfy the robots. The graph-based algorithm is complete and resolution optimal, which means that if a feasible path exists, the graph-based method can find an optimal path; otherwise, it will return failure. However, graph-based methods do not perform well in large-scale problems (e.g., industrial robotic arms) because the search space obtained by the graph-based discretization of the state space is too large.

The sampling-based planning method is another important type of planning algorithm. Instead of discretizing the state space, sampling-based planning is to construct a graph or tree by randomly sampling in the state space. Compared with graph-based planning algorithms, sampling-based planning algorithms perform better in large-scale problems.

The sampling-based planning algorithm is probabilistically complete; thus, the probability of finding a feasible path approaches 1 when the number of samples tends to infinity. Probabilistic Roadmaps (PRM) [6] and RRT [7] are two important algorithms of sampling-based planners. PRM is a multi-query motion planning algorithm, which obtains a graph representing spatial connectivity through random sampling in the state space, and then generates a feasible path through graph search. After constructing the graph, the PRM can be used to search multiple paths. However, it is a time-consuming process to construct a map of the entire space for a single search. By contrast, as a single-query motion planning algorithm, the RRT only explores the state space by growing a tree rooted at the start state and is thus faster than the PRM. The RRT algorithm involves three steps, which first randomly samples a state in state space, followed by selecting the nearest nodes of the random tree, and growing from the nearest neighbor sampling point to a random state. Once the tree reaches the goal region, the search is successful, and RRT will return to a feasible path.

Although RRT can quickly find an initial path in high-dimensional space, it still has many disadvantages. For example, owing to the random sampling, the variance of its search time is so large that it may require a long time to find a feasible path. The RRT does not perform well in narrow channel scenarios [8]. Moreover, the path found by RRT may be far away from the optimal path [9] since the path is randomly generated. Rapidly Exploring Random Tree Star (RRT*) [10] was considered an important extension of RRT. After finding an initial path, RRT* keeps optimizing the initial path by continuously sampling [11]. Compared with RRT, RRT* introduces two operations of neighbor search and rewiring tree to reach the optimal path. It can be proved that the path obtained by RRT* is optimal when the number of sampling approaches infinity. However, for RRT*, plenty of time and memory usage are required to reach the optimal path [12]. Similarly, RRT* also suffers from the problem of the large variance in search time.

Great efforts were devoted to improving the quality of the paths obtained by RRT and RRT*. For instance, by extending RRT* to kinodynamic systems, corresponding Kinodynamic RRT* can obtain an optimal trajectory that satisfies dynamic constraints. Anytime-RRT* can quickly re-plan the path with any point as the initial point. Alternatively, improving the search efficiency is another focus in RRT algorithm-related work, such as speeding up the search and minimizing the variance of search time. For instance, RRT-Connect [13] constructs two trees rooted in the initial state and the target state and makes the two trees grow toward each other. In [14], a 2D Gaussian mixture model is used to find a high-quality initial solution quickly. Batch Informed Trees [15] limits the state space to an incrementally increasing subset and thus quickly finds a feasible path. Unfortunately, these methods only performed well in certain environments.

Combining RRT with other path search algorithms is an important way to speed up the search process. In [16], the artificial potential field algorithm is incorporated in RRT* to accelerate convergence rate, but the planning time may dramatically increase in complex environments. The A*-RRT* [17] algorithm uses the path generated by the A* algorithm to guide the sampling procedure of the RRT* planner, which significantly accelerates the convergence speed. However, quite a long time is required for A* to find an initial path when it comes to large-scale problems. LM-RRT [18] applies reinforcement learning methods to guide the growth of trees, but the learning-based methods may not perform well in the new environment.

Heuristic-biased sampling is another important method. Informed RRT* [19] limits the sampling area to an ellipsoidal containing a feasible path, thereby returning an approximate optimal path faster than RRT*. Hybrid-RRT [20] combines RRT-Connect and Informed RRT* to accelerate the convergence speed of the algorithm. In [21], the obstacle boundary information is used to ignore bad sampling points. Dynamic-Domain RRTs [22] avoid the samples that are too far from the current tree. RRT*-Smart [23] uses the initial path to generate biasing sampling points.

Recently, learning-based methods have been widely applied in motion planning research. Neural RRT* [12] uses deep learning to learn a distribution probability for sampling. RL-RRT [24] explores deep reinforcement learning policy as a local planner and uses distance function that learns by deep learning to bias tree-growth towards the target area. In [25], a learning algorithm combining Inverse Reinforcement learning and RRT* is used to learn the RRT*'s cost function. The DL-P-RRT* algorithm applies deep learning to the RRT* algorithm using the virtual artificial potential field to learn the artificial potential field function. Learning-based methods perform well in some environments, but they may suffer from bad generalization ability in new environments.

In this paper, we propose an RRT-based planning method for optimal motion planning, called Fast-RRT. Compared with RRT*, Fast-RRT accelerates the search speed of the optimal path by order of magnitude. To be specific, Fast-RRT divides the search for the optimal path into two steps. The Improved-RRT algorithm is first used to find a feasible path quickly, and then the Fast-Optimal algorithm fuses multiple paths to obtain an optimal path.

For Improved RRT, two important improvements were made compared to the RRT algorithm, such as Fast Sampling and Random Steering. Fast Sampling improves the search speed of the RRT algorithm by refusing to sample in the explored area and solves the problem of the large variance in the search time of the RRT algorithm. Random Steering randomly chooses the direction to expand when the expansion fails, which solves the problem of poor performance of the RRT algorithm in narrow channel scenarios. By introducing these two improvements, the Improved-RRT algorithm can quickly find a feasible solution. Besides, Fast-Optimal can obtain the optimal path by virtue of fusing multiple paths and has a faster convergence speed as compared to RRT*. After a new feasible path is searched by Improved-RRT, Fast-Optimal merges it with the current optimal path to obtain a better path. The fusion operation of Fast-Optimal also consists of two steps, including path fusion and path fine-tuning. Path fusion can fuse multiple initial paths to obtain a better path, while path fine-tuning can quickly adjust the fusion path, which speeds up the search for the optimal path. Due to these advantaged characteristics, the search speed of Fast-RRT for finding a near-optimal path is 20 times faster than the RRT* algorithm. Therefore, our Fast-RRT algorithm exhibits great potential in practical motion planning applications.

The rest of this paper is organized as follows. The formal definition of the motion planning problem and the necessary background are presented in Section 2. In Section 3, the proposed Fast-RRT method in this paper is be defined. Section 4 shows the simulation and evaluation of our experimental results. At last, a brief summary and outlook is be presented in Section 5.

## 2. Background

In this section, we present the related backgrounds of this paper. The formal definition of motion planning problems is first introduced, followed by discussing the related algorithms such as RRT and RRT*.

### 2.1. Problem Definition

Two issues of motion planning, including the feasible solution and the optimal problems, is be discussed and defined in a similar way to [26].

Let $X \in \mathbb{R}^n$ be the state space of the planning problem, where $n \in \mathbb{N}$ is the dimension of the motion planning problem. $X_{obs} \in X$ is the obstacle space, which cannot be reached due to the collision with obstacles. $X_{free} = X/X_{obs}$ represents the free state space. $x_{init} \in X_{free}$ and $x_{goal} \in X_{free}$ are the initial state and the goal state, respectively, and $X_{goal} = \left\{ x \in X \middle| \|x - x_{goal}\| < r \right\}$ is the goal region. A feasible path is defined as a path $\sigma : [0 : T] \to X_{free}$ such that $\sigma(0) = x_{init}$ and $\sigma(T) \in X_{goal}$.

The feasible problem of motion planning is to find a feasible path if one exists; otherwise, it should report failure. Problem 1 defines the feasibility problem of path planning.

**Problem 1 (Feasible Path Planning)** Given a state space of planning problem $X \in \mathbb{R}^n$, a free space $X_{free}$, an initial state $x_{init}$ and a goal region $X_{goal} \in X_{free}$, find a path $\sigma : [0 : T] \rightarrow X_{free}$ such that $\sigma(0) = x_{init}$ and $\sigma(T) \in X_{goal}$, if one exists. If no feasible path exists, then report failure.

Path cost is an important concept in motion planning. By defining the cost function, a feasible path is assigned a non-negative cost. The optimal problem of motion planning is to find a path with minimal cost. Problem 2 formalizes the optimal problem of motion planning.

**Problem 2 (Optimal Path Planning)** Given a state space $X \in \mathbb{R}^n$, a free state space $X_{free}$, an initial state $x_{init}$ and a goal region $X_{goal} \in X_{free}$, if a solution to problem1 exists, find a path $\sigma : [0 : T] \rightarrow X_{free}$ such that $\sigma(0) = x_{init}$, and $\sigma(T) \in X_{goal}$ and $c(\sigma^*) = min_{\sigma \in \sum_{free}} c(\sigma)$.

The path-find algorithm based on sampling is a time-consuming process refereed to find an optimal path. Therefore, a near-optimal path is usually obtained through a certain number of iterations. The near-optimal problem refers to finding a path where the difference of cost and the optimal path is less than the threshold. Problem 3 formalizes the near-optimal problem of motion planning.

**Problem 3 (Near-Optimal Path Planning)** Given a state space $X \in \mathbb{R}^n$, a free state space $X_{free}$, an initial state $x_{init}$ and a goal region $X_{goal} \in X_{free}$, if a solution to problem 1 exists, and $\sigma^*$ is the solution to Problem 2, find a path $\sigma : [0 : T] \rightarrow X_{free}$ such that $\sigma(0) = x_{init}$, and $\sigma(T) \in X_{goal}$ and $c(\sigma) < c(\sigma^*) * (1 + threshold)$.

### 2.2. RRT and RRT*

The RRT algorithm is the basis of the proposed Fast-RRT, while RRT* is an important method to find the approximate optimal path. Therefore, the RRT and RRT* algorithms are introduced in Algorithm 1 and Algorithm 2, respectively.

The RRT is a single query search algorithm based on sampling, which can quickly find a feasible path. In the initialization phase, RRT builds a tree with the initial state $x_{init}$ as the node. In each iteration, RRT first randomly samples a state $x_{rand}$ in state space $X_{free}$ and select the nearest vertex $x_{near}$. Then, the RRT algorithm will generate $x_{new}$ through the steering function, as shown in Figure 1. If the edge of $\{x_{near}, x_{new}\}$ is obstacle-free, then $x_{new}$ will be added to the set of nodes, and $\{x_{near}, x_{new}\}$ will be added to the set of edges. If $x_{new}$ is located in the target area $X_{goal}$, the search is successful, and this result will be returned.

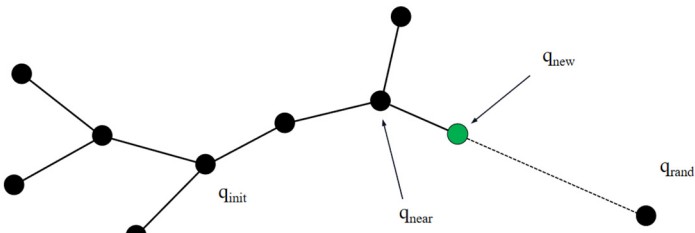

**Figure 1.** Schematic illustration of the expansion steps of the RRT algorithm. The RRT algorithm performs random sampling in the state space to obtain $q_{rand}$, and the vertex closest to $q_{rand}$ is calculated as $q_{near}$. Then, the random tree of RRT extends a certain distance from $q_{near}$ to $q_{rand}$ to obtain $q_{new}$.

The path found by RRT may be far away from the optimal path. RRT* overcomes this problem by introducing a rewire step. If the edge $\{x_{near}, x_{new}\}$ is obstacle-free, then nodes with a distance less than $r$ around $x_{new}$ will be calculated to determine the optimal parent node of $x_{new}$. In addition, RRT* not only adds $x_{new}$ to the tree, but also considers it as a replacement parent node for existing neighboring nodes. Therefore, as the sampling times approach infinity, RRT* continuously adjusts the random tree and finally finds an optimal

path. However, the RRT* is time-consuming and thus is not suitable for applications that need to find an optimal path quickly.

---

**Algorithm 1.** RRT

---

Input: $x_{init}$, $x_{goal}$ and *Map*
T.init()
**for** $i = 1 \ldots N$ **do**
    Let $x_{rand} \leftarrow UniformSample(Map)$;
    Let $x_{near} \leftarrow Nearest(Map)$;
    Let $x_{new} \leftarrow Steer(Map)$;
    Let $E_i \leftarrow Edge(x_{new}, x_{near})$
    If $ObstacleFree(Map, E_i)$
        T.addNode($x_{new}$)
        T.addEdge($E_i$)
            If $x_{new} \in X_{goal}$ then
                Success();
            End
    End
End

---

---

**Algorithm 2.** RRT*

---

Input: $x_{init}$, $x_{goal}$ and *Map*
T.init()
**for** $i = 1 \ldots N$ **do**
    Let $x_{rand} \leftarrow UniformSample(Map)$;
    Let $x_{near} \leftarrow Nearest(Map)$;
    Let $x_{new} \leftarrow Steer(Map)$;
    If $ObstacleFree(Map, E_i)$
        $X_{near} \leftarrow NearC(T, x_{new})$;
        $x_{min} \leftarrow ChooseParent(X_{near}, x_{near}, x_{new})$;
        $T.addNodeEdge(x_{min}, x_{new})$;
        $T.rewire()$;

---

## 3. Method

In this section, we introduce our Fast Rapidly-exploring Random Trees (Fast-RRT) algorithm. The framework of the proposed method is presented in Section 3.1, which consists of two modules, including Improved-RRT and Fast-Optimal. The details are introduced in Sections 3.2 and 3.3.

### 3.1. Framework of Fast-RRT

The detailed process of Fast RRT is shown in Algorithm 3, which includes two steps. The first step, called Improved-RRT, aims to find a feasible path, and the second step, called Fast-Optimal, aims to merge this feasible path with the current optimal path in order to obtain a near-optimal path.

For Improved RRT, two important improvements related to the sampling and the extended strategies were made compared to the baseline RRT algorithm, such as Fast Sampling and Random Steering. These strategies are introduced in Sections 3.2.1 and 3.2.2, respectively.

Fast-Optimal also consists of two parts, path fusion (Section 3.3.1) and path fine-tuning (Section 3.3.2). Path fusion can fuse multiple initial paths to obtain a better path, while path fine-tuning can quickly adjust the fusion path, which speeds up the search for the optimal path. Thus, our Fast-RRT can quickly find a near-optimal path by fusing multiple paths. By contrast, the RRT* adjusts the tree to find the near-optimal path, which is a time-consuming process. The Fast-Optimal algorithm is introduced in Section 3.3.

---

**Algorithm 3.** Fast-RRT Algorithm

---

**Input:** $x_{init}$, $x_{goal}$ and *Map*
**Output:** A path $T$ from $x_{init}$ to $x_{goal}$
**for** $i = 1 \ldots N$ **do**
    $T_{init} \leftarrow Improved\_RRT\left(x_{init}, X_{goal}, Map\right)$;
    **If** $T_{init}$ *Isnot None* **then**
        $T_{optimal} \leftarrow FastOptimal\left(T_{optimal}, T_{init}\right)$

---

### 3.2. Improved RRT

As an improved pathfinding algorithm, Improved-RRT introduces a fast sampling strategy and a random expansion strategy. The details of the proposed algorithm are shown in Algorithm 4. Fast Sampling improves the search speed of the RRT algorithm by refusing to sample in the explored area and solves the problem of the large variance in the search time of the RRT algorithm. Random Steering randomly chooses the direction to expand when the expansion fails, which solves the problem of poor performance of the RRT algorithm in narrow channel scenarios. By introducing these two improvements, the Improved-RRT algorithm can quickly find a feasible solution.

---

**Algorithm 4.** Improved RRT

---

**Input:** $x_{init}$, $X_{goal}$ and *Map*
**Output:** A path $P$ from $x_{init}$ to $x_{goal}$
T.init()
**for** $i = 1 \ldots N$ **do**
    $x_{rand} \leftarrow FastSample(Map)$;
    $x_{near} \leftarrow Near(T, x_{rand})$;
    $x_{new} \leftarrow RandomSteer(x_{nearest}, x_{rand})$
    $E_i \leftarrow Edge(x_{new}, x_{near})$
    **If** $ObstacleFree(Map, E_i)$ **then**
        $T.addNode(x_{new})$
        $T.addNode(E_i)$
        If $x_{new} \in X_{goal}$ then

---

#### 3.2.1. Fast Sampling

The basic RRT algorithm randomly samples a state in the entire state space and guides the tree to expand to it. When a new node is expanded by the RRT algorithm, the RRT calculates the distance $d$ between the node and the target point. If $d$ is less than the set threshold $r$, the algorithm then finds the target point until the end of searching; otherwise, the search continues. Therefore, every time a new node is expanded, the RRT algorithm detects the area with the node as the center and the distance from the center less than r. We defined this area as the explored area. The RRT algorithm can finally realize the exploration of all areas through continuous sampling and growth.

However, this sampling strategy may produce many invalid expansions and result in a large variance in the search time. An example of invalid expansion is shown in Figure 2. The blue areas in the figure indicate the explored area of three nodes. $x_{rand}$ is the state sampled by the RRT algorithm, $x_{new}$ is the state generated by steering operation, and then $\{x_{near}, x_{new}\}$ is added to the edge set. However, this expansion is invalid because the red exploration area of the newly generated node completely falls within the original exploration area and fails to expand the newly explored area.

To address the above issue, we propose a sampling algorithm called Fast-Sampling. By refusing to sample in the explored area, Fast-Sampling can avoid a large number of invalid growths. Figure 3 indicates the principle of Fast-Sampling, where a new vertex $x_{new}$ is added to the tree, its surrounding area $x_{aruond} = \left\{ x \in X \middle| \|x - x_{goal}\| < r \right\}$ is set as the explored area. As shown in Figure 3, the sampled state $x_{rand1}$ is located in the explored

area, the corresponding guided expansion is then invalid. The Fast-Sampling samples another random state $x_{rand2}$, which guides the random tree $T$ expand to an unexplored area. Compared with RRT, Fast-Sampling speeds up the exploration speed and reduces memory usage. Algorithm 5 presents the detail of Fast-Sampling.

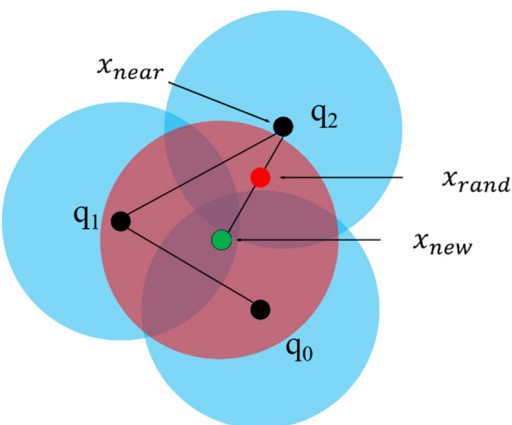

**Figure 2.** Schematic illustration of the invalid expansion caused by RRT's random sampling strategy. $x_{rand}$ is the randomly sampled point, $x_{\_near}$ is the nearest neighbor of $x_{rand}$, and $x_{new}$ is the expansion point. Unfortunately, this step of expansion is invalid because the random tree is not brought close to the target point or expanded to a new area.

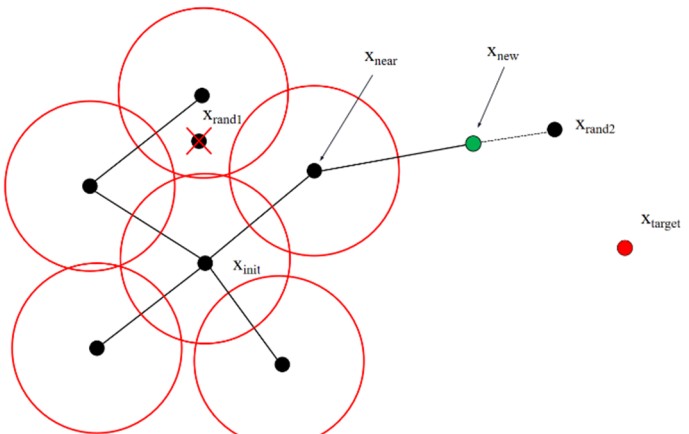

**Figure 3.** Schematic illustration of the Fast-Sampling sampling strategy. The red circle is the explored area. Since the first sampling point $x_{rand1}$ falls in the explored area, it is regarded as an invalid sampling. After re-sampling, the second sampling point $x_{rand2}$ is obtained. $x_{rand2}$ bootstrap tree expansion to obtain $x_{new}$, thus avoiding invalid expansion.

---

**Algorithm 5.** Fast Sampling

---

**Input:** $x_{rand}$
Let $x_{rand} \leftarrow UniformSample()$;
**While** $x_{rand} \in X_{explored}$ **do**
　　　　$x_{rand} \leftarrow UniformSample()$;

---

### 3.2.2. Random Steering

In addition to Fast-Sampling, the Random Steering strategy was also proposed to solve the problem of poor performance of the RRT algorithm in narrow channel scenarios. The inferior performance of the RRT algorithm can be attributed to the invalid expansion introduced in Section 3.2.1, and a large number of failed expansions caused by the narrow

passages are shown in Figure 4. In order to pass through the narrow passages, the tree $T$ should expand toward the right state space. While the expansion guided by $x_{rand}$ collides with an obstacle, resulting in the failed expansions.

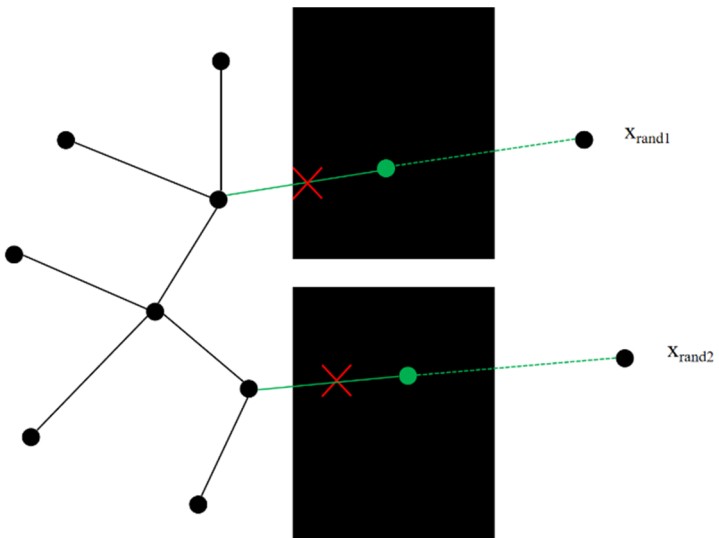

**Figure 4.** Schematic illustration of invalid expansions in narrow passages. $x_{rand1}$ and $x_{rand2}$ are two randomly sampling points. Since the random tree can only pass through a narrow passage, the expansions guided by $x_{rand1}$ and $x_{rand2}$ both collided with obstacles.

To avoid this problem, we adjusted the expansion algorithm and named it as Fast-Expand. As shown in Figure 5, if the expansion guided by $x_{rand1}$ fails, Fast-Expand will randomly generate a direction $\theta$ for expansion. If edge $(x_{near}, x_{new1})$ is obstacle-free, then they will be added to the tree; otherwise, another random state will be sampled by Fast-RRT. Fast-Expand enables the random tree of the Fast-RRT algorithm to pass through the narrow passage quickly. The details of the Fast-Expand algorithm are shown in Algorithm 6.

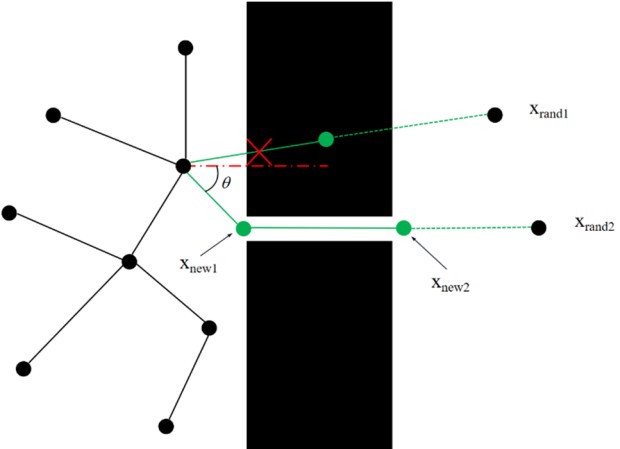

**Figure 5.** Schematic illustration of random expansion strategy. When the tree $T$ expands towards $x_{rand1}$, it randomly samples a direction $\theta$ to expand because it collides with an obstacle. In this way, the Fast-Expand samples another random tree and provides a successful pass through the narrow channel after getting $x_{rand2}$ in the next sampling.

**Algorithm 6.** Fast Expanding

**Input:** $Map$, $T$, $x_{rand}$
**Output:** $x_{new}$
Let $x_{near} \leftarrow Nearest(T, x_{rand})$;
Let $E \leftarrow Edge(x_{new}, x_{near})$;
**if** $ObstacleFree(E, Map)$ **then**
      **Return** $x_{new}$
**Else**
      $\theta \leftarrow Rand()$
      $x_{new} \leftarrow Expand(x_{near}, \theta)$

### 3.3. Fast-Optimal

Except for Improved RRT, Fast-Optimal is another important part of our Fast RRT algorithm. As an optimal pathfinding algorithm, Fast-Optimal is an asymptotically optimal algorithm that obtains a better path by fusing the new initial path sought with the current optimal paths. Compared with the RRT* algorithm, Fast-Optimal accelerates the convergence speed of the algorithm. The Fast-Optimal algorithm consists of two steps involving path fusion and path fine-tuning. Path fusion can fuse multiple initial paths to obtain a better path, while path fine-tuning can quickly adjust the fusion path, which speeds up the search for the optimal path. They are introduced in Sections 3.3.1 and 3.3.2, respectively.

#### 3.3.1. Path Fusion

Path fusion refers to intercepting a part of each of two paths to combine them into a better path. In this case, the evaluation index of the path is the path length. Therefore, path fusion is introduced to fuse multiple initial paths to obtain a better path with a shorter length than the two paths. Figure 6 illustrates the algorithm flow chart of path fusion. In each iteration, a random non-optimal path $P_{new}$ is obtained by the Imporved RRT algorithm. Due to the high efficiency of Fast-RRT, $P_{new}$ can be quickly obtained. Then $P_{new}$ is fused with the current optimal path $P_{optimal}$ to obtain a more optimal path, thus being the new $P_{optimal}$. By continuously generating new paths and fusing them with the current optimal path, the path fusion algorithm will result in the final optimal path.

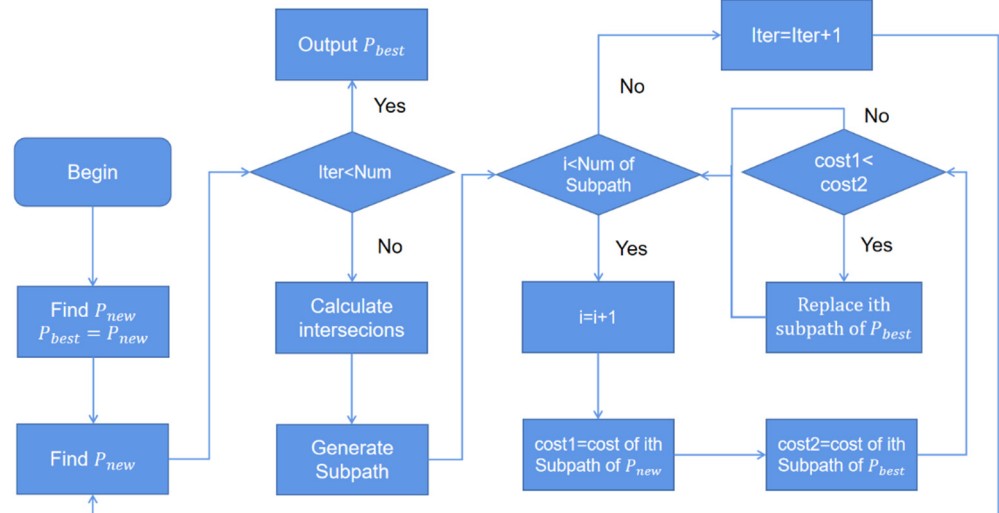

**Figure 6.** Schematic illustration of the algorithm flow chart of path fusion.

The process of path fusion is shown in Figure 7. It consists of the following steps: the first step is to find the intersection, and the second step is to select the sub-path. In the first step, the intersection points of paths $path_{new}$ and $path_{best}$ is be calculated and used to divide the path into multiple sub-paths. Given that the nodes of the two paths do not completely

overlap, if the distance between $point_1$ and $point_2$ is less than the set threshold $r$, they are considered to be overlapped and set the midpoint $point_{mid}$ as overlap point. In the second step, we select sub-paths from $path_{new}$ and $path_{obest}$ to connect each intersection. As shown in Figure 6, the sub-path of $P_{new}$ connecting $q_{start}$ and $q_{near}$ is shorter than the sub-path of $path_{best}$ connecting $q_{start}$ and $q_{new}$. Thus, the sub-path of $path_{best}$ is used to connect $q_{start}$ and $q_{near}$. Similarly, the sub-path of $path_{best}$ is used to connect $q_{near}$ and $q_{goal}$ because it is shorter than that of $path_{new}$. In this way, the two paths can be combined into a better path than the individual paths. By continuously fusing the current optimal path $path_{best}$ with the newly generated random path, the algorithm quickly obtains a near-optimal path. The details of the path fusion algorithm are shown in Algorithm 7.

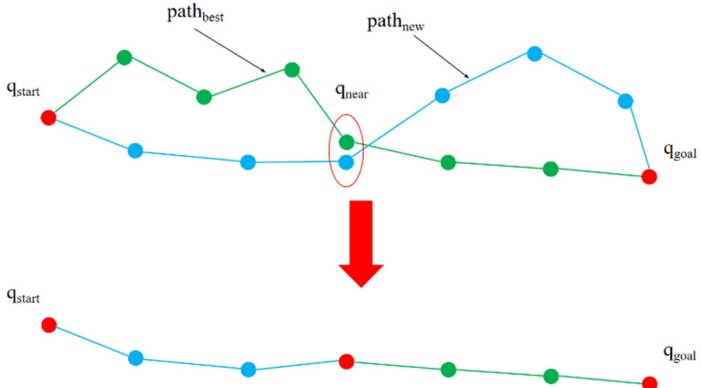

**Figure 7.** Schematic illustration of the path fusion process. The green path $path_{obest}$ is the currently obtained optimal path, the blue path $path_{new}$ is the newly obtained feasible path, and $q_{near}$ is the intersection of the two paths.

---

**Algorithm 7.** Path fusion

---

**Input:** $P_{best}$, $P_{new}$
**Output:** $P_{best}$
**Foreach** $point_{new}$ $of$ $P_{new}$ **do**
        **Foreach** $point_{best}$ $of$ $P_{best}$ **do**
                **If** $d = \|point_{new}, point_{best}\| < Thr$ **then**
                        intersections $append$ $(point_{new}.index(), point_{best}.index())$
**For** $k = 1 : size(joints) - 1$ **do**
        $(start_{new}, start_{best}) = intersections(i)$
        $(end_{new}, end_{best}) = intersections(i)$
        $Path_{opt} \leftarrow SubPath(P_{best}, start_{new}, end_{new})$
        $Path_{new} \leftarrow SubPath(P_{new}, start_{best}, end_{best})$
        **If** $Cost(Path_{opt}) < Cost(Path_{new})$ **then**
                $P_{best}[start_{best} : end_{best}] \leftarrow P_{new}[start_{new} : end_{new}]$

---

### 3.3.2. Path Fine-Tuning

In order to speed up the path optimization, we also propose a path fine-tuning strategy that obtains a better path by simply adjusting the path obtained by the path-fusion algorithm. The process of the path fine-tuning is shown in Figure 8. The path composed of green points is obtained by the path-fusion algorithm, in which $q_{near1}$, $q_{near2}$ and $q_{near3}$ are the coincident points of $P_{new}$ and $P_{optimal}$, respectively.

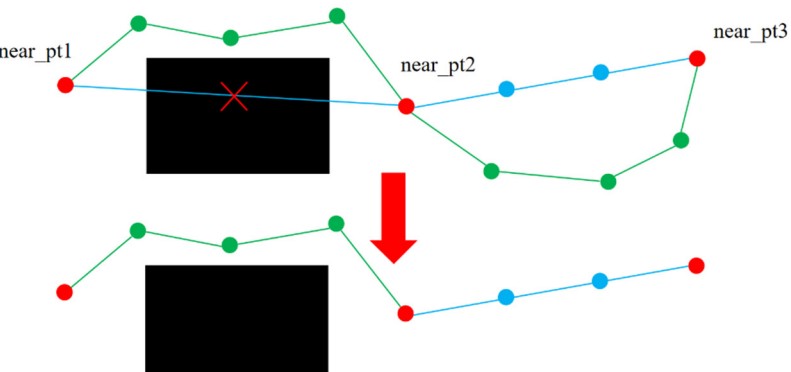

**Figure 8.** Schematic illustration of the process of path fine-tuning. *point*1, *point*2, *point*3 are the intersection points of path fusion. Since the connection between *point*1 and *point*2 collides with obstacles, the original path is retained. In contrast, the connection between *point*2 and *point*3 is unobstructed so that the original path can be replaced by a new straight path.

During the process of path fine-tuning, we try to connect each intersection with a straight line. The connection between $q_{near1}$ and $q_{near2}$ collides with the obstacle; thus, the original path is saved. In contrast, the connection between $q_{near2}$ and $q_{near3}$ does not collide with obstacles, so we can use straight to connect these two points because the direct connection is shorter than the original polyline connection. Therefore, the path fine-tuning algorithm can obtain a better path with few turns, which is more convenient for the robots to execute. The details of fast fine-tuning are shown in Algorithm 8.

---

**Algorithm 8.** Path fine-tuning

---

**Input:** $P$, *intersections*, *Map*
**Output:** $P_{tuning}$
Let $P_{tuning} = []$
**For** $k = 1 : size(intersections) - 1$ **do**
      $idx1 \leftarrow near\_list[k]$
      $point1 \leftarrow P[idx1]$
      $idx2 \leftarrow near\_list[k+1]$
      $point2 \leftarrow P[idx2]$
      **if** $ObstacleFree(point1, point2)$ **then**
            $Subpath \leftarrow Generate(point1, point2)$
            $P_{tuning}$ append $Subpath$
      **else**
            $Subpath \leftarrow P(idx1 : idx2)$
            $P_{tuning}$ append $Subpath$

---

## 4. Simulation and Result

In order to evaluate the performance of the proposed algorithm, a series of experiments are performed. As mentioned above, the Fast-RRT includes two modules, Improved RRT and Fast-Optimal. The Improved RRT algorithm can quickly find the initial path, while the Fast-Optimal algorithm can obtain a near-optimal path by combining the initial paths. Therefore, we first compared Improved-RRT with RRT to prove its efficiency in finding the initial solution. Then the comparison between Fast-Optimal and RRT is shown to prove the efficiency of the algorithm in finding the near-optimal solution.

Specifically, three experiments are designed to evaluate the efficiency of the proposed algorithm. The first experiment is to find a feasible path in a complex environment. The simulation environment is shown in Figure 9, which contains multiple obstacles of different shapes. The second experiment is to find a feasible path in a narrow passage scene, which is difficult for RRT. The simulation environment containing multiple narrow passages is shown in Figure 10. The third experiment is to find a near-optimal path by using the same

environment as the first experiment. All experiments were run on a desktop computer, configured with Intel Core i7-7700k processor, 16 GB RAM, Windows 10, and Matlab R202.

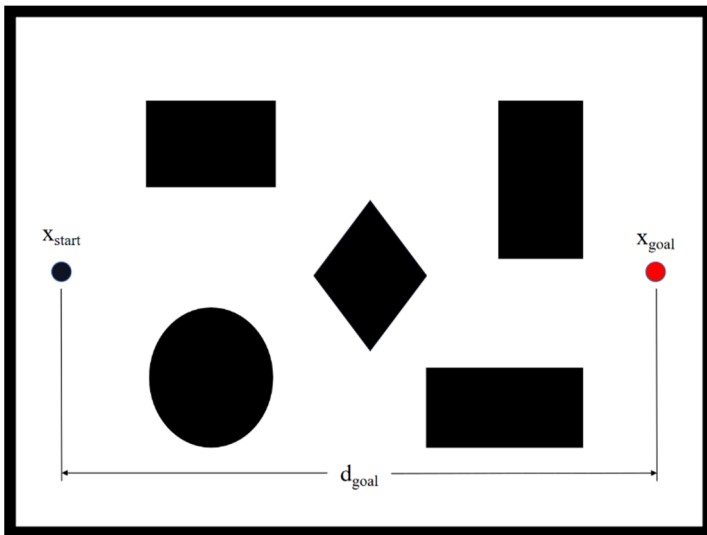

**Figure 9.** Schematic illustration of simulation environment 1 containing multiple obstacles of different shapes. The distance from the state $x_{start}$ to the target state $x_{goal}$ is $d_{goal}$, and its value is 1000.

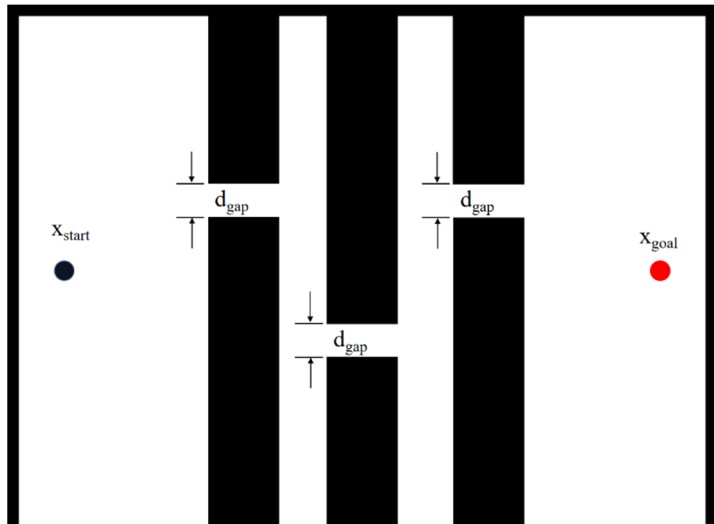

**Figure 10.** Schematic illustration of simulation environment 2, where three narrow passages are required from the starting state $x_{start}$ to the goal state $x_{goal}$. The distance from the state $x_{start}$ to the target state $x_{goal}$ is $d_{goal}$. The value is set to 1000.

### 4.1. Find Initial Path

This experiment is to test the algorithm's ability to find an initial solution, which refers to finding an obstacle-free path from $x_{start}$ to $x_{goal}$. The RRT algorithm is used as a benchmark algorithm to verify the efficiency of the Improved-RRT algorithm. The simulation environment used for this experiment is shown in Figure 8. The length and width of the environment are 1200 and 900, respectively. $d_{goal}$ is the distance from the initial state $x_{start}$ to the goal state $x_{goal}$, and its value is set to 1000. Different step sizes affect the complexity of the problem. In our case, we designed multiple sets of experiments and set the step sizes to 10, 20, 30, 40, and 50, respectively. As the step size decreases, the searching number required to find a feasible solution increase along with the complexity of the problem.

The results of the Fast-RRT algorithm and RRT algorithm in a test are shown in Figure 11. The red points are the random sampling state of the algorithm, the green line is the random tree generated by the algorithm, and the blue line is the feasible path obtained by the search. Obviously, the sampling points of the RRT algorithm are randomly distributed at any position in the state space. By contrast, the sampling points of the Fast-RRT algorithm are sparsely distributed around the random tree and densely distributed in the area far from the random tree. Fast-RRT avoids sampling in the area that the random tree has reached, which is beneficial for guiding the random tree to expand to the unknown area. At the same time, compared with the RRT algorithm, Fast-RRT uses fewer nodes to cover the space due to the presence of the abundant random trees of the RRT algorithm, resulting in more invalid extensions. Finally, Fast-RRT has multiple random edges near the boundary of the obstacle. Although it increases memory consumption, the Fast-RRT facilitates the random tree to bypass the obstacles to reach the target point quickly.

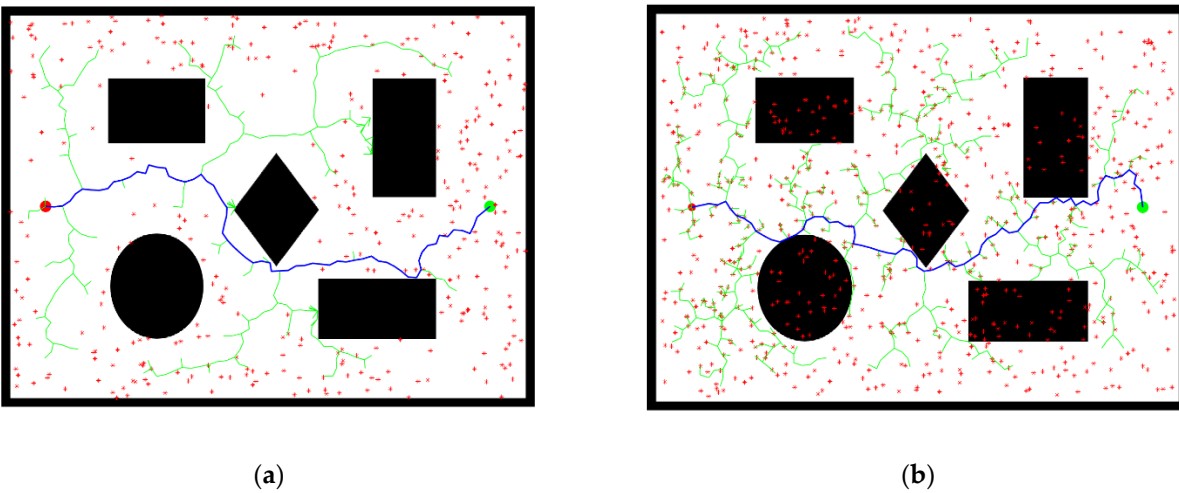

(**a**)　　　　　　　　　　　　　　　　　　　　　　　　　　(**b**)

**Figure 11.** Comparison of the operation results of (**a**) the improved RRT algorithm and (**b**) the RRT algorithm. The sampling points of the Improved RRT algorithm are more densely distributed in the area where the random tree has not reached, and its random tree is more efficient to achieve the state space. As for the RRT, the sampling points are evenly distributed throughout the space. Compared with Improved RRT, it uses more nodes to achieve coverage of the state space.

Finally, the efficiency of the algorithm was evaluated by search time and memory consumption. Specifically, the number of nodes in the random tree is used to evaluate memory consumption. The average and variance are used as two indicators for the characterization of the search time. The average value of the search time represents the average performance of the algorithm, while the variance indicates its stability, which is both very important for practical applications (e.g., robots). In our case, RRT and Fast-RRT were run 100 times, and then the running time and the number of nodes of the random tree were recorded. The running time of the two algorithms and the average and variance of the number of nodes were further calculated.

Figure 12 shows the results of the running time. Compared with the RRT algorithm, the average and variance of the search time of the Fast-RRT algorithm are significantly reduced. When the step size is set to 50, the average and variance of the search time are 0.016 s and 0.009 s, respectively. By contrast, for the RRT algorithm, the average and variance of the search time are 0.043 s and 0.043 s, respectively. Correspondingly, the average search time and the variance within the Fast-RRT algorithm are only 37.2% and 20.9% of the RRT algorithm, respectively. As the step size decreases, the complexity of the problem increases along with the gap between them. When the step size is set to 10, the average and variance of the search time of the Fast-RRT algorithm are only 23.5% and 8.3% of the RRT algorithm.

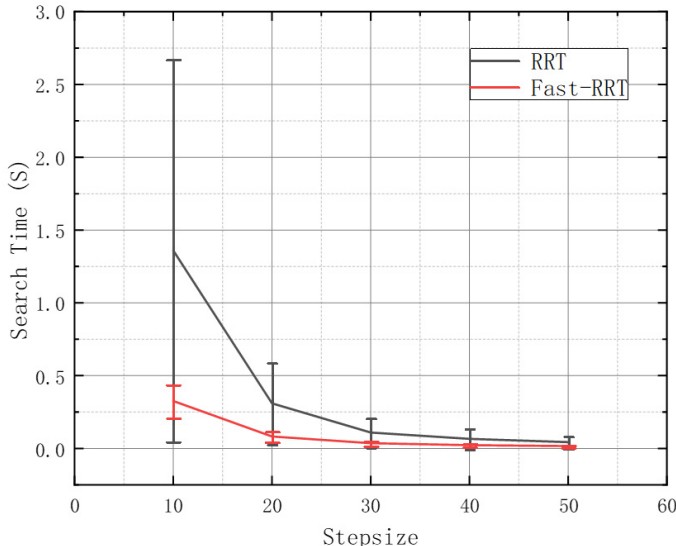

**Figure 12.** Comparison of the search time between RRT algorithm (black) and Improved-RRT algorithm (red) with different step sizes.

Figure 13 compares the results of the memory consumption of RRT and Fast-RRT algorithms. Compared with the RRT algorithm, Fast-RRT also shows remarkable advantages in memory usage. At step sizes of 10, 20, 30, 40, and 50, the average number of nodes of the random tree is 659.6, 322.9, 207.9, 158.4, and 129.5, respectively. In contrast, the RRT algorithm has a larger number of nodes of the random tree of 1182.3, 551.9, 323.6, 244.5, and 198.8, respectively. At the same time, the variance of the number of nodes in the Fast-RRT algorithm random tree is also smaller. When the step size is set to 10, the variance of the number of nodes in the Fast-RRT random tree (113.9) is only 22.1% of the RRT algorithm (515.6).

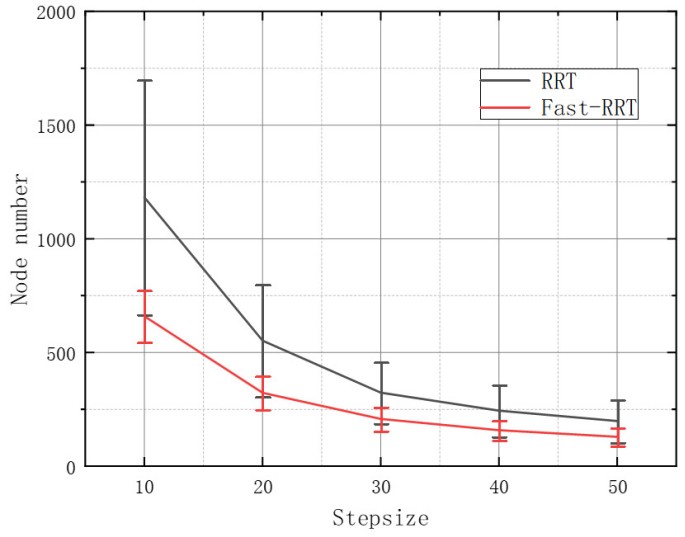

**Figure 13.** Comparison of the node number of the random tree in the Fast-RRT algorithm and the RRT algorithm with different step sizes.

*4.2. Narrow Passages Scene*

The ability of our proposed Fast-RRT algorithm is also evaluated to find a feasible path in an environment with narrow passages. The algorithm needs to find an obstacle-free path from $x_{start}$ to $x_{goal}$. Due to the requirement of a large number of sampling, the RRT algorithm does not perform well in making the random tree pass through these narrow

channels. The environment, in this case, is shown in Figure 9. The length and width of the environment are 1200 and 900, respectively. There are three narrow passages in the environment, and the width of the channel is set to 80. $x_{start}$ is the start state and $x_{goal}$ is the goal state. $d_{goal}$ is the distance from $x_{start}$ to $x_{goal}$, which is set to 1000.

The average and variance of the search time are used to evaluate the efficiency of the Improved RRT algorithm, and the RRT algorithm is used as the benchmark algorithm. Several groups of experiments were performed with the step sizes set to 10, 20, 30, 40, and 50, respectively. For each experiment, we ran the RRT algorithm and the Improved-RRT algorithm 100 times, respectively. Then the average and variance of the search time of these algorithms were calculated, and obtained results are shown in Figure 14.

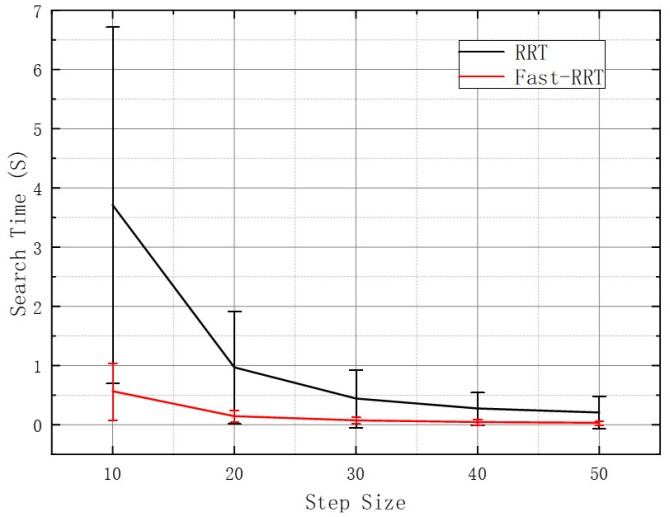

**Figure 14.** Comparison of the search time between Fast-RRT algorithm and RRT algorithm in the narrow passages as a function of different step sizes.

The results show that the average and variance of the search time within the Improved-RRT algorithm are significantly smaller than those of the RRT algorithm. When the step size is set to 10, 20, 30, 40, and 50, the average search time of the Improved-RRT algorithm is 0.567 s, 0.146 s, 0.076 s, 0.047 s, and 0.035 s, respectively. The values are 15.3%, 15.1%, 17.1%, 17.0%, 16.8% of the RRT algorithm, respectively. Moreover, the variance of search time of the Improved-RRT algorithm is 0.480 s, 0.101 s, 0.056 s, 0.050 s, and 0.032 s, which are 16.5%, 15.9%, 10.5%, 11.5%, 11.7% of the RRT algorithm, respectively. Therefore, the Improved-RRT algorithm can find a feasible path faster than the RRT algorithm and has better stability. Compared with the results of Section 3.1, the difference between the average and variance of the search time of these two algorithms is further increased, indicating that the random expansion strategy proposed in this paper has a significant effect on the pathfinding problem of narrow passage environment.

*4.3. Find Near-Optimal Path*

The algorithm's ability to find a near-optimal path is also measured. The algorithm needs to find an obstacle-free path from $x_{start}$ to $x_{goal}$, whose length differs from the optimal path's length is less than threshold $r$. The environment used in this experiment is the same as Section 3.1, the environment is shown in Figure 8, the length of the environment is 1200, the width of the environment is 900, and $d_{goal}$ is 1000. For this environment, the length of the optimal path is 1035.

Similarly, the average and variance of the search time are applied to evaluate the efficiency of the algorithm. The differences between the length of the near-optimal path and the length of the optimal path are set to 5%, 10%, 15%, 20%, and 25%, respectively. At a threshold of 5%, we need to find a path with a length of less than 1087, as the length of the optimal path is 1035. The RRT* is used as a control algorithm to verify the efficiency of the

proposed algorithm. The step size is also set to 30 as the same as the Fast-RRT algorithm. In each experiment, we ran the Fast-RRT algorithm and the RRT* algorithm 100 times and recorded the running time. Finally, the average and variance of the search time of these two algorithms were calculated. The final results are presented in Figure 15.

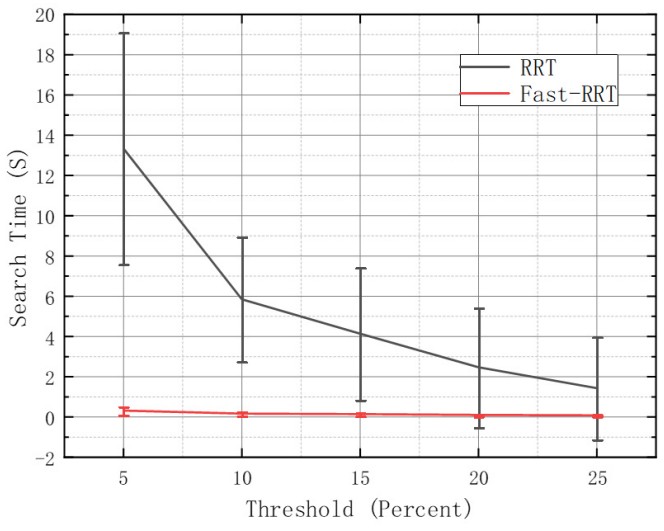

**Figure 15.** Comparison of the time required for Fast-RRT and RRT* algorithms to find an approximate optimal path when the gap between the approximate optimal path and the theoretical optimal path is different thresholds. It can be seen that compared with the RRT* algorithm, the Fast-RRT algorithm has an order of magnitude advantage in search time.

As expected, the average and variance of the search time within the Fast-RRT algorithm is smaller than those of the RRT* algorithm. At a threshold of 5%, the RRT* algorithm shows an average and variance of the search time of 13.34 s and 5.75 s, respectively. Impressively, the average and variance of the search time of the Fast-RRT algorithm are 0.322 s and 0.207 s, respectively, which is only 2.4% and 3.6% of the RRT* algorithm. When the threshold is set up to 10%, 15%, 20%, and 25%, the average search time of the Fast-RRT algorithm is 2.9%, 3.4%, 4.0%, and 6.1% of the RRT* algorithm, respectively. Therefore, the search speed of the Fast-RRT algorithm is 20 times faster than the RRT* algorithm. Moreover, the variance of search time within the Fast-RRT algorithm is 3.4%, 2.4%, 2.0%, and 2.1% of the RRT* algorithm, demonstrating enhanced stability.

Thanks to the superior performance mentioned above, our Fast-RRT possesses great potential in the design and navigation of robots. After using sensors such as lasers and depth cameras to build an environment map, our Fast-RRT algorithm can be used to find a feasible path from the starting point to the target point. Moreover, due to the advantages of fast search speed and small search time variance, our Fast-RRT can be further applied to UAV navigation and other path search tasks that require high real-time performance.

## 5. Conclusions

In summary, we proposed a new RRT-based path planning algorithm, Fast-RRT, to improve the speed and stability of finding the initial path. Therefore, two improvements, such as sampling in the unexplored space and random expansion, were performed. Furthermore, a new algorithm for finding a new near-optimal path was further proposed to obtain a near-optimal path by combining and adjusting multiple feasible paths.

Compared with the RRT and RRT* algorithms, the proposed that Fast-RRT possesses remarkable advantages in speed and stability. For instance, within the Fast-RRT algorithm, the average search time and its variance to find a feasible path significantly reduced compared to the RRT algorithm. At the same time, the search speed of Fast-RRT for finding a near-optimal path is 20 times faster than the RRT* algorithm. Therefore, our Fast-RRT

algorithm exhibits great potential in practical motion planning applications. To further improve the performance of our algorithm, the combination of the Fast-RRT algorithm and other algorithms, such as Bidirectional RRT, Kinodynamic RRT*, and Information RRT*, will also be investigated in the future. At the same time, the application scenarios of our Fast RRT algorithm will also be extended from two-dimensional to multi-dimensional, as well as in actual motion planning tasks in future work.

**Author Contributions:** Conceptualization, Z.W. (Zhenping Wu) and Z.M.; methodology, Z.W. (Zhenping Wu); software, Z.W. (Zhenping Wu) and W.Z.; validation, Z.W. (Zhenping Wu), Z.M. and W.Z.; formal analysis, Z.W. (Zhenping Wu); investigation, Z.M.; resources, Z.M.; data curation, W.Z.; writing—original draft preparation, Z.W. (Zhenping Wu); writing—review and editing, Z.W. (Zhenping Wu) and Z.M.; visualization, W.Z.; supervision, Z.M. and Z.W. (Zhe Wu); project administration, Z.W. (Zhe Wu); funding acquisition, Z.M. All authors have read and agreed to the published version of the manuscript.

**Funding:** This work was supported by the National Natural Science Foundation (NSF) of China (No.61976014).

**Institutional Review Board Statement:** Not applicable.

**Informed Consent Statement:** Not applicable.

**Data Availability Statement:** The data presented in this paper are available on request from the first author.

**Conflicts of Interest:** The authors declare no conflict of interest.

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
