# Peer review of "Fast-RRT: A RRT-Based Optimal Path Finding Method"

_applsci, doi:10.3390/app112411777_

Round 1

Reviewer 1 Report

Overview.

The paper attempts to improve "Rapidly Exploring Random Trees(RRT)" performance in terms of speed, memory efficiency and finding the optimal path in facing obstacles. The authors tried to break down their proposed method into two modules, first finding a feasible path, and then fusing multiple paths to obtain an optimal path. 

Strengths.

The introduction is well-plotted. It first starts by explaining the problem and then it reviews the previous papers. The authors narrowed down the story in each paragraph to finally introduce their algorithm. 

Weaknesses.

The idea is not very well explained, I believe the authors should spend more time describing their method.

The literature review needs work since several recent papers on this topic are not cited in this paper.

The paper mentioned the superiority of their algorithm in terms of time and memory consumption, so a very detailed theoretical and experimental evaluation was expected. 

The paper needs to map each element of the framework and describe them in more detail. 

From what I could understand, the Fast-RRT consists of two modules, Improved RRT for finding the feasible path and Fast Optimal for merging the previously found paths. However, it was hard to keep track of each sub-sections. 

The paper also needs proofreading. 

General comments.

Based on the proposed structure of Fast-RRT, the Improved-RRT module should be executed multiple times and then the result should be sent to the next step (time complexity). 

This step requires a time complexity evaluation. 

In Section 3.1, how does the Fast-RRT algorithm discriminate between the optimal and near the optimal path? "By contrast, Fast-RRT can quickly find a near-optimal path". Examples are needed here. 

Fast sampling: it is not clear what is the exact purpose of this subsection. The authors should design a better plan to discuss each part of their algorithm. 

Section 3.2.1: "This sampling strategy may produce many invalid expansions and result in a large variance in the search time." What is the metric for evaluating the validity of the expansion? It would be good to have at least a reference here.

Figure 2. doesn't provide intuition about why the generated path is invalid. If it is invalid in terms of Euclidean distance, the authors need to add some notion to clarify this point.  

Fast sampling: Regarding the idea of the explored area, how does the model identify the explored area? What radius from the generated point is called "explored area"?

There is no relation between "Random Steering" and the previous topic. The authors are required to keep the consistency between each subsection.

Path Fusion. 

"Path fusion refers to intercepting a part of each of two paths to combine them into a better path". What is the "better path"? How do you measure the goodness of a path? 

The process of Path Fusion algorithms is not clear from Figure 6. It would have been better if the authors added a flowchart or more notions in the Figure 6 to explain the algorithm. Putting all steps of an important algorithm in one long paragraph reduces the readability and understandability of the algorithm.

Experimental results. 

The authors claim that the time and memory complexity of their method is much less than the RRT algorithm. The authors had to consider these points for the execution of experiments:

1- Testing the Fast RRT against other models - only comparing the Fast RRT with Baseline RRT makes the analysis limited.

2- Statistically proving the significance of their algorithm

3- Designing a few more experiments to prove the significance of the model

Author Response

Dear Reviewers,

Thanks a lot for reviewing our manuscript “Fast-RRT: A RRT-based Optimal Path Finding Method”. The comments and suggestions are all valuable and very helpful for improving the quality of our paper. We have studied your comments carefully and substantially revised our manuscript. The revisions were addressed point by point below, and the involved changes have been marked up using the “Track Changes” function in the revised manuscript.

Point 1: The idea is not very well explained, I believe the authors should spend more time describing their method.

The paper needs to map each element of the framework and describe them in more detail. From what I could understand, the Fast-RRT consists of two modules, Improved RRT for finding the feasible path and Fast Optimal for merging the previously found paths. However, it was hard to keep track of each sub-sections. 

Response 1: We apology for the insufficient expression that cause doubts about each element of the framework and the function of each part of the algorithm. In this work, we propose an algorithm to quickly find the optimal path called Fast-RRT. The Fast-RRT algorithm consists of two modules including Improved RRT (introduced in Section 3.2) and Fast-Optiml (introduced in Section 3.3). The Improved RRT is aim to quickly find an initial path, and the Fast-Optimal is to merge multiple initial paths to obtain a near-optimal path.

For Improved RRT, two important improvements have been made as compared to the RRT algorithm, such as Fast Sampling (Section 3.2.1) and Random Steering (Section 3.2.2). Fast Sampling improves the search speed of the RRT algorithm by refusing to sample in the explored area, and solves the problem of large variance in the search time of the RRT algorithm. Random Steering randomly chooses the direction to expand when the expansion fails, which solves the problem of poor performance of the RRT algorithm in narrow channel scenarios. By introducing these two improvements, the Improved-RRT algorithm can quickly find a feasible solution.

Fast-Optimal also consists of two parts, Path fusion (Section 3.3.1) and Path fine-tuning (Section 3.3.2). Path fusion can fuse multiple initial paths to obtain a better path, while Path fine-tuning can quickly adjust the fusion path, which speeds up the search for the optimal path.

To clarify these and make our paper easier to understand, we strengthened the relationship between each part by introducing the framework and function of each element in our algorithm, and provided more details in the revised manuscript. 

Point 2: The literature review needs work since several recent papers on this topic are not cited in this paper.

Response 2: Thanks for the suggestion. The recent progress on the related topic has been introduced and cited in the revised manuscript. Taking kinodynamic RRT* and anytime RRT* as examples, the former can obtain a path that satisfies dynamic constraints, and the latter can quickly realize re-planning. As an improvement of RRT, these two methods are widely used in practical engineering.

Point 3: The paper mentioned the superiority of their algorithm in terms of time and memory consumption, so a very detailed theoretical and experimental evaluation was expected.

Based on the proposed structure of Fast-RRT, the Improved-RRT module should be executed multiple times and then the result should be sent to the next step (time complexity). This step requires a time complexity evaluation. 

Response 3: Thanks for the suggestion. In our case, the search time of the experimental part and the mean and variance of the number of nodes are introduced to prove the efficiency of the algorithm, instead of performing theoretical analysis related to time complexity. It is well-known that sampling-based algorithms such as RRT are probabilistically complete, which means that when the number of sampling approaches infinity, the probability of obtaining a feasible solution will approaches 1. Therefore, the time complexity of the algorithm is theoretically infinite and is not convenient to analyze it. In this article, we conduct multiple experiments and count the average and variance of the calculation time and memory consumption of different algorithms. The experimental results show that the average and variance of the search time of our algorithm are significantly smaller than RRT and RRT*. We believe that this result has been able to prove that our algorithm has obvious advantages in efficiency.

Point 4: The paper also needs proofreading. 

Response 4: Thanks for the suggestion. We have carefully revised our manuscript again and corrected the writing errors.

Point 5: In Section 3.1, how does the Fast-RRT algorithm discriminate between the optimal and near the optimal path? "By contrast, Fast-RRT can quickly find a near-optimal path". Examples are needed here. 

Response 5: Within Fast-RRT algorithm, each path  corresponds to a loss function value . The optimal path  refers to the path with the smallest loss function value, that is, for any path , there is . The near-optimal path refers to the difference in the loss of the path  and the optimal path  due to the set threshold, . In our algorithm, our path loss function is the length of the path. Assuming that the length of the optimal path is 1000 and the threshold is set to be 5%, then the near-optimal path refers to the path with the length of less than 1050.

Besides, in order to explain this concept in detail, we defined Problem3 in Section 2 to explain the problem of finding a near-optimal path definition and near-optimal path planning.

Point 6: Fast sampling: it is not clear what is the exact purpose of this subsection. The authors should design a better plan to discuss each part of their algorithm. 

Response 6: Thanks for the suggestion. Fast-sampling is a part of Improved RRT with aims to guide the random tree to grow to the unexplored area by sampling in the unexplored area, thereby speeding up the search and reducing the variance of the search time. To clarify this, we have revised the Method part and described the framework of our algorithm carefully so that readers can better understand the role of each part.

Point 7: Section 3.2.1: "This sampling strategy may produce many invalid expansions and result in a large variance in the search time." What is the metric for evaluating the validity of the expansion? It would be good to have at least a reference here.

Figure 2. doesn't provide intuition about why the generated path is invalid. If it is invalid in terms of Euclidean distance, the authors need to add some notion to clarify this point.  

Response 7: Thanks for the suggestion. When the RRT algorithm expands a new node, it will detect whether the distance between the node and the target point is less than the set threshold or not. If the node does not reach the target area, there is no target point in the area where the distance from the node is less than the threshold. This area is called the explored area. The RRT algorithm guides the growth of a random tree through sampling until the explored area gradually covers the state space, and finally covers the target point. By contrast, for the base RRT algorithm, as shown in Figure 2, the sampling point for the growth of the guide tree falls in the explored area. In that case, the expanded node will not increase the explored area of the RRT tree. Such a sampling point and the related growth of the random tree guided by it are invalid.

To clarify it, the related discussion has been included and Figure 2 has been modified in the revised manuscript.

Point 8: Fast sampling: Regarding the idea of the explored area, how does the model identify the explored area? What radius from the generated point is called "explored area"?

Response 8: As mentioned above, when the RRT algorithm expands a new node, it will detect whether the distance between the node and the target point is less than the set threshold or not. If the node does not reach the target area, there is no target point in the area where the distance from the node is less than the threshold. This area is called the explored area. For a two-dimensional environment, the explored area corresponding to the node is a circle.

Point 9: There is no relation between "Random Steering" and the previous topic. The authors are required to keep the consistency between each subsection.

Response 9: Thanks for the comment. Actually, “Random steering” is another important part of our Improved RRT algorithm. Improved-RRT consists of two parts, Fast-Sampling and Random steering. Fast-Sampling is used to improve the sampling efficiency, while Random steering is to overcome the problem of low efficiency of the RRT algorithm in narrow channel scenes. To clarify this, the introduction of these two parts and their relationship with Improved RRT have been added in the revised manuscript for better understanding.

Point 10: "Path fusion refers to intercepting a part of each of two paths to combine them into a better path". What is the "better path"? How do you measure the goodness of a path? 

Response 10: Thanks for the comment. The quality of a path is determined by its evaluation function. In our case, the evaluation function is the length of the path. Therefore, a path with the shorter length is called a better path.

Point 11: The process of Path Fusion algorithms is not clear from Figure 6. It would have been better if the authors added a flowchart or more notions in the Figure 6 to explain the algorithm. Putting all steps of an important algorithm in one long paragraph reduces the readability and understandability of the algorithm.

Response 11: Thanks for the valuable suggestion. In the revised manuscript, the overall process of Path Fusion has been described more carefully and a flowchart for the path fusion algorithm has been added in Figure 6 to better explain the algorithm.

Point 12: 1- Testing the Fast RRT against other models - only comparing the Fast RRT with Baseline RRT makes the analysis limited.

Response 12: Thanks for the comment. If I understood correctly, the reviewer suggested that more experiments to compare our Fast RRT algorithms with other methods, such as Connect-RRT and Information RRT*, should be set up to further verify the efficiency of our algorithm. Essentially, our proposed Fast RRT is a great improvement in sampling and growth of RRT, which can also be easily combined with other algorithms such as Connect-RRT, to improve their sampling and expansion efficiency, thus obtaining faster search speed. Therefore, our algorithm and Connect-RRT algorithm are not in a competitive relationship, but can cooperate to achieve better results. Based on these factors, we only designed a comparison between our Fast RRT and the RRT and RRT* algorithms to prove that our algorithm has better performance and can replace RRT and RRT* in combination with other improvement strategies. We believe that the current experiment is sufficient.

Point 13: 2- Statistically proving the significance of their algorithm.

Response 13: Thanks for the suggestion. In our case, we conducted 100 experiments on our Fast RRT, RRT and RRT* algorithms, respectively, and compared the search time, the number of nodes, and the average and variance of the path length. Among them, the average can prove the efficiency of our algorithm, the variance can demonstrate the stability of our algorithm. On the basis of these two values, we believe that our algorithm has obvious advantages over RRT and RRT*.

Point 14: Designing a few more experiments to prove the significance of the model

Response 14: Thanks for the suggestion. In the manuscript, we designed the search of feasible solutions in a multi-obstacle environment, including three experiments such as the search of feasible solutions in a multi-obstacle environment, the search of feasible solutions in a narrow passage environment, and the search of the optimal solution in a multi-obstacle environment. These experiments proved that our algorithm has obvious advantages over RRT and RRT* in searching for feasible solutions and optimal solutions in different environments. For sure, more experiments can better verify the effectiveness of our algorithm. Unfortunately, due to the limited revised time, we cannot design and complete new supplement experiments in the short term. We will keep your suggestion in mind for the ongoing works.

Reviewer 2 Report

Dear authors, 

I don't have many comments regarding your contributions and presentation.

You have minor spelling mistakes, so reread your paper, just in case.

I was wondering only about the experiments. 

So basically, you performed experiments in two environments and compared with two methods the speed and the node numbers, and you reached much better results. Do you think that number of experiments is sufficient? I understand that implementation of different algorithms requires a lot of work, but are there some other methods you can maybe compare with? 

Also, you may write something about requirements for applying your algorithm in real world (obtaining the map of a room etc). What are the possible adaptations to 3D?

Author Response

Dear Reviewers,

Thanks a lot for reviewing our manuscript “Fast-RRT: A RRT-based Optimal Path Finding Method”. The comments and suggestions are all valuable and very helpful for improving the quality of our paper. We have studied your comments carefully and substantially revised our manuscript. The revisions were addressed point by point below, and the involved changes have been marked up using the “Track Changes” function in the revised manuscript.

Point 1: So basically, you performed experiments in two environments and compared with two methods the speed and the node numbers, and you reached much better results. Do you think that number of experiments is sufficient? I understand that implementation of different algorithms requires a lot of work, but are there some other methods you can maybe compare with? 

Response 1: Thanks for the comment. If I understood correctly, the reviewer suggested that more experiments to compare our Fast RRT algorithms with other methods, such as Connect-RRT and Information RRT*, should be set up to further verify the efficiency of our algorithm. Essentially, our proposed Fast RRT is a great improvement in sampling and growth of RRT, which can also be easily combined with other algorithms such as Connect-RRT, to improve their sampling and expansion efficiency, thus obtaining faster search speed. Therefore, our algorithm and Connect-RRT algorithm are not in a competitive relationship, but can cooperate to achieve better results. Based on these factors, we only designed a comparison between our Fast RRT and the RRT and RRT* algorithms to prove that our algorithm has better performance and can replace RRT and RRT* in combination with other improvement strategies. We believe that the current experiment is sufficient.

Point 2: Also, you may write something about requirements for applying your algorithm in real world (obtaining the map of a room etc).

Response 2: Thanks for the suggestion. We agree that the application of our algorithm in the real world is very important. For our Fast RRT, its superior performance enables the application in the design and navigation of robots. For example, after using sensors such as lasers and depth cameras to build an environment map, we can use our Fast-RRT algorithm to find a feasible path from the starting point to the target point. The related discussion has been added in Section 4.3 as highlighted in the revised manuscript.

Point 3: What are the possible adaptations to 3D?

Response 3: Thanks for the suggestion. In the manuscript, we only discussed the 2D situation because the 2D environment is more intuitive and can be better used to introduce our algorithm. In fact, our algorithm can also be simply applied to a 3D environment only if changed to sampling in the three-dimensional space, and then changed the explored area from a circular range to a spherical range. In future work, we will apply our algorithm to a multi-dimensional environment. The related discussion has been included in Section 5 in the revised manuscript.
